# A Machine-Learning Method to Assess Growth Patterns in Plants of the Family Lemnaceae

**DOI:** 10.3390/plants11151910

**Published:** 2022-07-23

**Authors:** Leone Ermes Romano, Maurizio Iovane, Luigi Gennaro Izzo, Giovanna Aronne

**Affiliations:** Department of Agricultural Sciences, University of Naples Federico II, 80055 Portici, Italy; maurizio.iovane@unina.it (M.I.); luigigennaro.izzo@unina.it (L.G.I.); aronne@unina.it (G.A.)

**Keywords:** duckweed, machine learning, image analysis, machine training, aquatic plants, Lemnaceae, Lemna

## Abstract

Numerous new technologies have been implemented in image analysis methods that help researchers draw scientific conclusions from biological phenomena. Plants of the family Lemnaceae (duckweeds) are the smallest flowering plants in the world, and biometric measurements of single plants and their growth rate are highly challenging. Although the use of software for digital image analysis has changed the way scientists extract phenomenological data (also for studies on duckweeds), the procedure is often not wholly automated and sometimes relies on the intervention of a human operator. Such a constraint can limit the objectivity of the measurements and generally slows down the time required to produce scientific data. Herein lies the need to implement image analysis software with artificial intelligence that can substitute the human operator. In this paper, we present a new method to study the growth rates of the plants of the Lemnaceae family based on the application of machine-learning procedures to digital image analysis. The method is compared to existing analogical and computer-operated procedures. The results showed that our method drastically reduces the time consumption of the human operator while retaining a high correlation in the growth rates measured with other procedures. As expected, machine-learning methods applied to digital image analysis can overcome the constraints of measuring growth rates of very small plants and might help duckweeds gain worldwide attention thanks to their strong nutritional qualities and biological plasticity.

## 1. Introduction

Image analysis has changed the way scientists experiment in numerous fields [1]. The image analysis approach allows scientists to frame time-specific data that can be analysed later. This methodology has been adopted in multiple plant science research fields [2]. Image analysis software is the go-to technology to correctly satisfy the needs of modern research data. In several areas of study (including genetics), the requirement of image analysis software that could quantify tiny differences among plants’ phenotypes has been mandatory and has led us to enter the so-called “big data era” in plant science [3,4]. Thanks to the breach made by the genetic field, this software analysis method soon became mandatory in numerous other areas, such as botany, agronomy, and forestry [5,6,7,8,9,10].

Within the “big data era”, scientists are now facing the new challenge of extrapolating scientific-sounding data from the monstrous amount produced by image analysis [11,12]. This represents the birth of the “artificial intelligence era” [13], in which computer intelligence substitutes humans to extrapolate scientific-quality data among large quantities. The advent of artificial intelligence in plant science is already paying off [14]. In numerous fields of plant science, this technology is speeding up the process and excluding numerous errors made by human operators [11,15,16,17,18]. Although artificial-intelligence interfaces are still too complicated for most plant biologists, some software leads to the better use of this technology in numerous research fields [14,19]. Among others, ilastik^®^ is a supervised machine-learning software (learning from training data) that brings machine-learning-based image analysis to end-users without extensive computational expertise [20].

Ilastik^®^ provides end-users with a supervised machine-learning experience without requiring extensive training data. This is achieved thanks to the accurate machine-training feature of the software that can fine-tune training via a “paint”-like interface [21]. Ilastik^®^ contains predefined workflows that can be used for image segmentation, object classification, counting, and tracking [20]. Moreover, a specific setup of the machine-training ilastik^®^ process can be reutilised numerous times, and applying a particular feature of the program, “batch analysis”, can be performed theoretically with an infinite number of images [21].

This paper proposes the use of ilastik^®^ in a low-cost setup aimed at getting a new standardised method to perform image analysis of the aquatic plant family Lemnaceae. These hydrophytes have been often mentioned referring to their small size and fast growth [22,23,24]. However, their current appreciation is moving toward these plants’ exceptional nutritional qualities [23]. Additionally, Lemnaceae are gaining worldwide attention in numerous other fields, such as phytoremediation, plant science, biomonitoring, and closed bioregenerative systems [25,26,27]. Due to the simplicity of these biological systems, numerous scientists are evaluating the possibility of using these plants as a model [28]. Research in all these fields is constrained by the extremely small size of individuals that prevents applying the methods that are commonly used for biometric measurements and plant growth rates in all other flowering species. We suggest a new image analysis method via machine learning to boost knowledge and standardise the scientific analysis of the Lemnaceae plant’s growth. This approach can increase confidence in experimental results and speed up image analysis techniques by offloading the image analysis process to a machine [20]. Due to scientists’ strong interest in this family of plants, we shared the view that it is mandatory to standardise analysis methods [29] and decided to contribute to achieving such a goal.

In particular, we focused on methods able to identify fine changes in growth phenological processes more effectively than those based on the number of fronds used in the past [30]. The new method had to be applied in any growth-related tests, such as bioassay and laboratory tests, for Lemnacae and other floating aquatic plants.

More specifically, our work aimed to validate the utilisation of Ilastik^®^ software in monitoring the growth rate of Lemnaceae. Our approach was to highlight the possible effects of two light treatments on plant growth by applying the previously used analyses and the newly proposed method.

## 2. Results

To evaluate the use of the machine-learning system, we cultivated Lemna minor (9440) under different light-quality treatments. Moreover, we studied the growth rates with different methods. More specifically, the standard gold method has been defined by the ISO 20079 protocol. This method requires counting the number of fronds over a period of time at constant time intervals. Furthermore, two digital methods were also investigated, one previously described by Haffner et al. [29] and the newly described ilastik^®^ method. The three methods used the Naumann et al. [30] equation to calculate growth rates. The three methods’ results were compared to appreciate any existing differences. Additionally, the Ilastik^®^ method’s results were compared with those produced in Fiji.

Plants cultivated under the two different combinations of light conditions (white and white + red) grew healthy, with no visual sign of overall differences. Images of plants at the beginning and end of the experiment were used to calculate the relative growth rates (RGR) of Lemna minor by applying the three different methods (the ISO 20079 frond number evaluation, the Fiji image analysis software, and the ilastik^®^ machine-learning method). The ANOVA results showed no significant difference (*p* = 0.985) in the mean growth rates calculated with the three methods (Figure 1). Therefore, they are equally valid in calculating the growth rate of duckweeds.

Unlike the ISO 20079 method, the other two allowed us to calculate plant growth throughout the experiment by analysing a series of photos taken at regular intervals. We used these data to compare the two computerised methods further. Data showed a high correlation between the measurements by ilastik^®^ and Fiji methods, as represented in the scatter diagram (Figure 2). The strong correlation is supported by calculated coefficients of 0.99 for the Pearson coefficient and 0.99 for the Spearman coefficient. More specifically, the Pearson coefficient showed an almost perfect strength of agreement among data compared, while Spearman’s rho coefficient of rank correlation is 0.995. The 95% confidence interval ranges from 0.993 to 0.998. The conclusion is that there is a significant relationship between the two variables.

Due to the presence of outliers to the median line, we compared results by means of difference. The Bland–Altman (B&A) analysis is reported in Figure 3.

The results from the Heteroskedascity test with the White method have a *p*-value of 0.06; we fail to reject the null hypothesis and conclude that residuals show a homoscedastic distribution.

The three statistical analyses performed show that the two analysis methods can be used interchangeably.

### Time to Data

The previous paragraph shows that the newly described method is perfectly coherent with the results produced with the previously described method (Fiji) regarding data outcomes. We now consider the possible benefits and advantages in terms of time to produce data. 

The time required to run the analysis with both software by the same operator was 95.4 s for Fiji software and 300 s for ilastik^®^ per picture. The main difference between the two methods was that the operator who wanted to run the additional analysis with Fiji needed to start over again. This required the same amount of time per analysis (Figure 4). Differently, the operator that trained the machine by using ilastik^®^ to analyse the first image needed a time longer than that for one image with Fiji, but the operator could immediately run any batch analysis with no additional time required because the machine performed the same task for any number of pictures selected (Figure 4).

Furthermore, in the case of ilastik^®^, it was possible to save the machine-training parameters to be applied to possible future pictures taken under identical conditions (light, distance from the camera, camera setup, etc.).

According to the data recorded during the tests, the predictive analysis of time necessary to measure plant growth as a function of the number of images is in perfect accordance with the time model described by Formulas (1) and (2). Equation (1) describes plotted data from the Fiji software:(1)$$y_{1}=95.4x$$
where *y* refers to the time (in seconds) required by the operator to perform the analysis and *x* is the number of images to be analysed. Equation (2) describes the plotted data from the analysis conducted with the ilastik^®^ software:(2)$$y_{2}=300$$

## 3. Discussion

In this paper, we have demonstrated how a newly described method can be effective in calculating the biological effect of utilising machine learning during image analysis. Plants grown with different light recipes have not shown different growing patterns by means of relative growth rates differently from other studies conducted on crop species [31,32]. Furthermore, the data showed a strong correlation between our newly designed method with the older one (Fiji). Such a strong correlation maximises confidence in the new adoption of the method. Moreover, we have demonstrated how the presence of outliers has been warded off by testing for heteroskedasticity.

It is important to remark on the importance of RGR calculation in the field of study of the Lemancaeae plants as one of the few growth-monitoring tools. Upon having compared the RGR outcomes obtained with the different methods, we can conclude that the three methods are equally valuable for studying the growth rate of duckweed plants.

Nevertheless, the methods relying on image analysis do not require destructive measurements and can facilitate other in vivo analyses such as genetics [2]. 

The newly proposed approach of using a machine-learning system has numerous advantages and fewer disadvantages than previously proposed methods based on image analysis [29]. In fact, with the initial setup of a photo booth box, researchers can rely on coherent methods that discard human input in the analysis process. However, the newly described method has been applied in laboratory conditions, performing image analysis of plants from one and not multiple species. Further studies could provide helpful insight into its applications in open-field scenarios to evaluate the growth parameters of different species of Lemanaceae in the same photo. In this framework, researchers might utilise technologies that can picture entire ponds (drone photography) and define a single area to be analysed through ilastik^®^ software. This would be helpful in monitoring growth rate over time in raceway ponds for the massive production of Lemnaceae species [33].

It is important to remark that the value of the so-far-used methods remain not lowered; however, thanks to the more substantial presence of open-source software and more available technologies, tweaking these systems to researchers’ needs has become more accessible.

Our method is faster and as reliable as the other methods previously used to measure the growth rate of Lemnaceae [34,35]; however, our experiment did not compare results with a fresh or dry weight of plants because this was not the main aim of our work, and we considered reliable the correlation between weight and frond area [35]. The main advantage of applying our method is that it drastically reduces the time required by the operator to analyse the growth rate in Lemnaceae to only the time needed to train the machine. Noticeably, the latter corresponds to the time usually required to analyse a few images with the so-far-used image analysis methods. Moreover, by applying the ilastik^®^ procedure to different experiments designed to use the same photographic conditions and identical clones, the saved machine training can be stored by the researchers and used, theoretically, an infinite number of times. In these cases, our machine-learning approach simplifies the analysis methods to the “click of a button”.

Overall, both our result and what has been previously reported in the literature [29] confirmed the urge to use computer image processing to speed up the innovation process in Lemnaceae. As previously mentioned by other authors [29], the usage of this technology with low-cost hardware can define new qualitative standards in determining the growth rate of Lemnaceae.

## 4. Materials and Methods

Plants of *Lemna minor* were grown under identical environmental conditions through temperature, nutrient media, and background light conditions. The advent of light-emitting-diode (LED) technology allows scientists to provide plants with the exact amount and quality of light needed to maximise growth and efficiency. We experimented with different light recipes to validate the machine-learning method. More specifically, half of the plant samples were treated with a background light (white) and the other half with the integration of red light (white + red). Details of the cultivation and experiment setup, as well as of the data analyses, are reported below.

### 4.1. Plant Cultivation

*Lemna minor* (9440) was cultivated for 168h in a controlled-temperature chamber FOC 200IL Velp scientifica^®^ (Monza, at a constant temperature of 25 ± 0.5 °C. Five plants with two or more fronds were cultivated in a 150 mL glass beaker with 100 mL of Murashige and shook growth medium (Sigma-Aldrich—Murashige and Skoog basal medium, St. Louis, MO, USA) (pH adjusted to 5.8). The beakers were covered with a Petri dish to avoid water evaporation. The growth chamber was illuminated with a background white LED light.

Pictures of the growing plants were taken every 24 h with a Sony^®^ alpha 7 II camera equipped with a Sigma^®^ 50 mm Art F1.4, mounted on a fixed stand. Photos were shot under an illuminated photo booth with a white background to guarantee optimal sample illumination and contrast. Furthermore, camera photo parameters were kept constant throughout the experiment. 

### 4.2. Photo Booth Setup

To maintain a constant photo-shooting environment, we set up a photo booth in a dark room of our laboratory. This approach guarantees stable light conditions and centring the samples to the camera frame. We achieved this by buying a photo booth online and a camera tripod. Both components were fixed to a table to keep the camera distance and centring constant throughout the experiment. 

### 4.3. Light Quality and Quantity

We opted for a different light-quality setup to stimulate differences in growth; we decided to use the following light treatment setup. Plants were exposed to the same white background light. The existing difference among samples was due to providing extra monochromatic lighting to the red treatment. More specifically, single 3w red-coloured LEDs (not branded) were installed to achieve light treatment. Light quality and quantity are described in the following table. They are expressed as averages among the three replicas per treatment. Light quality and quantity were measured with a spectroradiometer (SS-110, Apogee Instruments Inc., Logan, UT, USA) to control the emission spectrum of each light treatment Table 1.

### 4.4. Measuring Systems

In this study, we adopted three different methods to evaluate the relative growth rates of *Lemna minor* during the experiment. As described by the ISO 20079, we used frond number as an evaluation method for growth during the investigation [30]. The other two approaches were achieved via computer software (Fiji and ilastik^®^ (Figure 5)) installed on Lenovo E480, intel CORE i7 8th gen, 16 GB of memory (8 GB minimum required by the software). Both methods produced quantitative information on the area occupied by the plant (in pixel). 

A plant’s growth rate can be calculated from the area or number of fronds described by Haffner et al. [29]. We employed Fiji software as defined by Haffner et al. [29]. The ilastik^®^ software was used following the protocol described in Appendix A. When training the machine, we started from the pictures where the frond number was the highest to better train the machine in understanding the picture composition. The output file from the software and the feature selected for the first picture was saved and could be used for other analyses. The three measuring systems were used to calculate the relative growth rate (RGR) described by Naumann et al. [30]. 

### 4.5. Statistical Analysis

Statistical analysis was conducted following four-step phases; first, we compared the three methods’ relative growth rates with the formula described by Naumann et al. [30]. In this phase, the outcome for the six growth rates compared to performing a one-way ANOVA analysis was performed with the SPSS Statistics 27 software (IBM Inc., Armonk, NY, USA). The ANOVA was fundamental in confirming that the three methods’ growth rates were in accordance. More specifically, the two computed methods were in accordance with the gold standard defined by the ISO 20079 protocol. Subsequently, we compared the proposed method (ilastik^®^ software) with the previously described one (Fiji). Agreement among the two computed methods (Fiji and ilastik^®^) was shown by calculating correlation coefficients with Pearson and Spearman. As described by Mcbride, the correlation can be defined as almost perfect because the value of ρc is 0.999 in the range >0.99 [36].

Furthermore, we compared utilising differences between the two measurement techniques with the Bland–Altman plot to underline the presence of bias between the two methods. As described by Dogan [29], Bland and Altman’s limits of agreement (LoA) have conventionally been used in medical research to evaluate the agreement between two methods of measurement for quantitative variables [37]. Nevertheless, Bland and Altman’s LoA method may be misleading in the presence of heteroskedastic distributions [38]. Due to the fact of outliers in the Bland and Altman graphic representation [39], we opted to test heteroskedasticity with the White test because it can better perform in the presence of nonlinear forms of heteroskedasticity (presence of outliers). 

### 4.6. Time Analysis of the Software Utilisation

The following formula was utilised to compare the time usage of the two software:(3)y=a+bx
where y is the time required by the operator to perform analysis with the software under evaluation, a is the time to set up the analysis with the given software. The letter b indicates the time for the operator to analyse a single image, and x is the number of images analysed. 

Consequently, the equation can be solved, respectively, for Fiji (4) and ilastik^®^ (5).
(4)$$y_{1}=bx$$
(5)$$y_{2}=a$$

To validate what was modelled by the mathematical equations, we provided quantitative data about the time to analyse the two computed methods. 

### 4.7. Relative Growth Rate 

Growth was calculated following the growth rate calculation, Ziegler et al. 2014 [40].
(6)$$RGR=(\ln X_{tn}-\ln X_{t0})/\left(t_{n}-t_{0}\right)$$
where *X* is the pixel in the area as described by Haffner et al. [29] and *t*_0_ and *t_n_* represent the start and the end of the test, respectively.

## 5. Conclusions

We presented a novel method designed to rapidly and inexpensively quantify the Lemnaceae growth rate by tracking frond surface variance at different time intervals. We showed that machine-learning technology can substitute traditional methods and document biological phenomena if well applied to the observed system. The proposed approach helps the researcher train the machine ad hoc to the specific requirements. This customisable method can be used across different Lemnaceae applications and other surface-floating aquatic plants.

It is important to remark on the importance of RGR calculation in the field of study of the Lemancaeae plants as one of the few monitors for growth. Upon having compared the RGR outcomes obtained with the different methods, we can conclude that the three methods are equally valuable for studying the growth rate of duckweed plants.

Nevertheless, the methods relying on image analysis do not require destructive measurements and can facilitate other in vivo analyses such as genetics [2].

The newly proposed approach of using a machine-learning system has numerous advantages and fewer disadvantages than previously proposed methods based on image analysis [29]. In fact, with the initial setup of a photo booth box, researchers can rely on coherent methods that discard human input in the analysis process. It is important to remark that the value of the old methods remains not lowered; however, thanks to the more substantial presence of open-source software and more available technologies, tweaking these systems to researchers’ needs has become more accessible.

## Figures and Tables

**Figure 1 plants-11-01910-f001:**
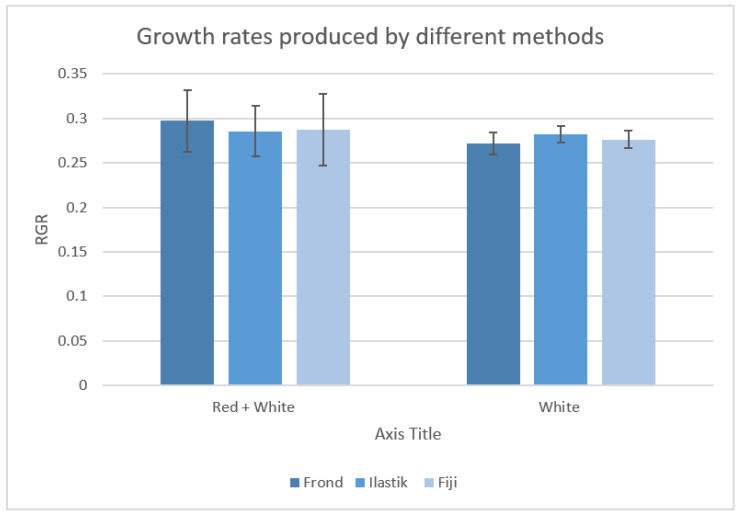
Graph compares RGR calculated with three methods (frond number, ilastik^®^ and Fiji) for the two experiment setups (red-light treatment and control).

**Figure 2 plants-11-01910-f002:**
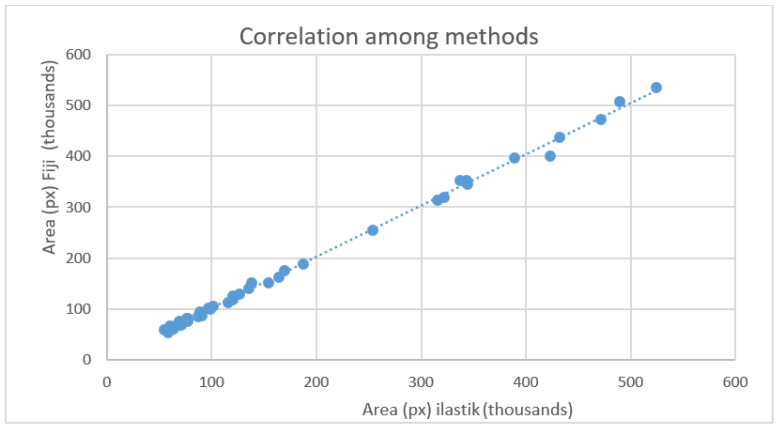
Graph shows plotted output data (pixel) of the same images analysed with the two methods (Fiji and ilastik^®^). As demonstrated by the data visualisation, a strong correlation between the two methods is present.

**Figure 3 plants-11-01910-f003:**
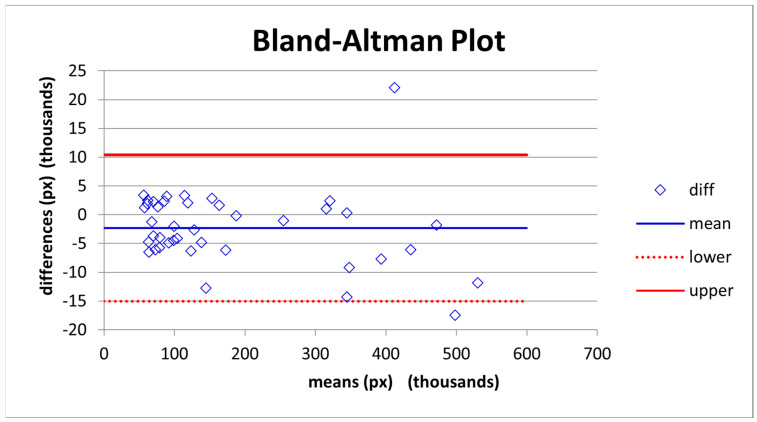
The B&A plot can be evaluated as in good agreement according to the scatter dispersion. The scattering of points is reduced, and points lie relatively close to the line representing mean bias. It is essential to consider the big numerical difference existing among data; this difference in the two outlier cases might (outside the limits of agreement) be due to human error during the Fiji analysis [3].

**Figure 4 plants-11-01910-f004:**
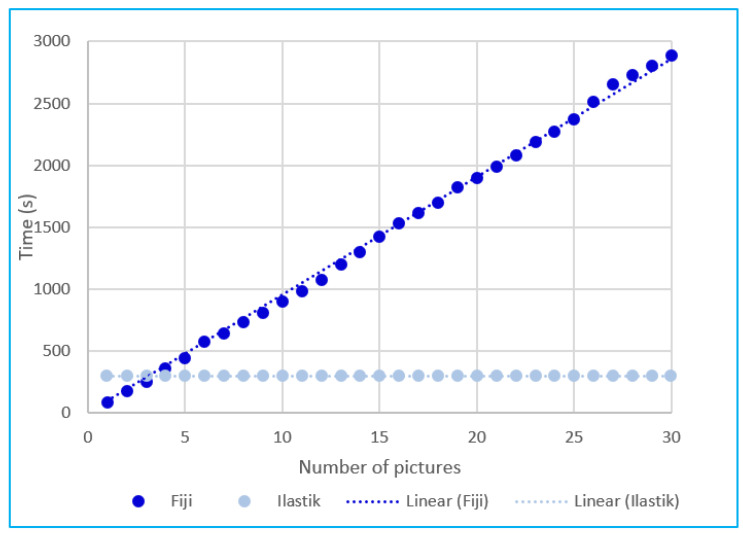
The time required by the operator to analyse the area occupied by fronds of Lemnaceae. The dark blue line represents the time required by the operator by employing the Fiji software; the light blue line represents the time requirements with the ilastik^®^ software. The x-axis represents the number of pictures; the y-axis represents the time requirement per picture.

**Figure 5 plants-11-01910-f005:**
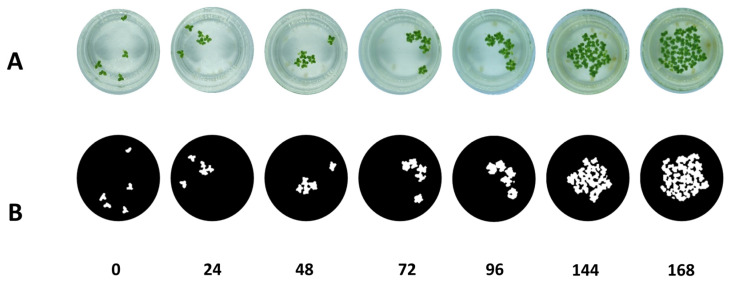
(**A**) pictures of *Lemna minor* at different time intervals (**B**) pictures processed by the ilastik^®^ software.

**Table 1 plants-11-01910-t001:** Total photon flux density (PFD) (μmol·s^−1^), photosynthetic photon flux (PPF) (μmol·m^−2^·s^−1^), yield photon flux (YPF) (μmol·s^−1^), and photosynthetic photon efficacy (PPE) (PPF umol/watts); R/FR is the red (R) light relative to the amount of far-red (FR) light.

	Total PFD	Stdev.	PPF	Stdev.	YPF	Stdev.	PPE	Stdev.	R/FR	Stdev.
**Red treatment**	**128.34**	**1.1**	**126.55**	**1.07**	**110.72**	**0.94**	**0.88**	**0**	**10.62**	**0.03**
**Control**	**129.69**	**0.98**	**126.53**	**0.61**	**107.06**	**0.66**	**0.83**	**0.02**	**7.08**	**0.01**

## Data Availability

Not applicable.

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
