# Peer review of "A Machine-Learning Method to Assess Growth Patterns in Plants of the Family Lemnaceae"

_plants, 2022, doi:10.3390/plants11151910_

Round 1
Reviewer 1 Report
The authors compared standard methodologies to quantify the area of duckweed from picture with a new method that involves machine learning. The authors observed that the new method requires more time to analyse the first pictures, but, after that, an infinite number of other pictures can be analysed without spending any extra time. If more than one picture needs to be analysed, the new method is extremely quicker than the other two methodologies. The authors did not detect any significant differences between the results obtained with the old methods and those obtained with the new method and they concluded that the latter can be recommended to save time.
The new method proposed is very interesting and I believe it helps to reduce time for biometric analyses of duckweed and, in general, of leaves of any species.
In order to improve the quality of the manuscript, the authors could explain better the steps described in the supplementary material. For example, in step 5, Feature selection: what αx and αy are? Many of the following steps are also not described properly and it is difficult for the operator to understand what he's doing. If there is an online training tool, it should be mentioned in the paper.
In the discussion you clearly stated that the method is as good as the traditional ones, just less time consuming. Can you, with this method, teach the machine to distinguish between species? In the past I analysed pictures in polycultures. I used Imagej and I could select the areas of different species separately to measure the respective areas. Can you do it with Ilastik? Also, how the software performs with pictures taken in the field? If it has limitations, they should be acknowledged in the discussion.
Please find below a few smaller comments:
Line 73: replace ‘shared’ with ‘suggest’?
Line 86: delete ‘have’
Line 174: ‘in the new adoption of the method.’ Please move ‘new’ before ‘method’
4.2 Photo booth setup 232 : please specify the distance between camera and duckweed mat
Line 256: ISO 20079 please, add year (2005) and include it in the list of references
Line 270: replace is with was
Author Response
Dear Reviewer,
Thank you for the revision process that you have conducted; we have accepted all the suggestions that you have proposed. Furthermore, we have provided further links in other to clear misinformation about the steps needed to complete the feature selection process. Since we have not developed the Ilastik software, we preferred to provide an external link to avoid falling into wrong definitions.
We have provided additional information on the possible use of ilastik in field scenarios. Unfortunately, multispecies discrimination is not feasible at the moment.
All minor comments have been modified accordingly.
Thank you
Best regards
Leone Ermes Romano

Reviewer 2 Report
In the current manuscript, Romano et al. have focused on methods able to identify fine changes in growth phenological processes more effectively than those based on the number of fronds used in the past. Although the topic is attractive, there are some concerns that should be addressed.
L 44: please provide some examples for "numerous fields of plant science". I listed some of them below. you can add more fields and references.
plant breeding (https://doi.org/10.1016/j.isci.2020.101890), in vitro culture (https://doi.org/10.1007/s00253-020-10888-2), stress phenotyping (https://doi.org/10.1016/j.tplants.2015.10.015), and system biology (https://doi.org/10.1007/s00253-022-11963-6).
L 169-167: Discussion should be improved. Indeed, the authors repeated the results. In the discussion part, the authors should discuss and compare their results, not repeat them.
Author Response
Dear Reviewer,
Thank you for the revision process that you have conducted; we have accepted all the suggestions that you have proposed. Furthermore, we have included all the suggested references.
Additionally, we have improved the discussion section to meet your suggestions.
Thank you
Best regards
Leone Ermes Romano

Round 2
Reviewer 2 Report
All my comments have been addressed. I think that the current form of the manuscript can be published in Plants.
Author Response
We have addressed your suggestions and made the appropriate modifications to the manuscript.